# Assessing the impact of obesity interventions in the early years: a systematic review of UK-based studies

Semina Michalopoulou [1], Maria Sifaki,[2] Jessica Packer,[1] Julie Lanigan [1], Claire Stansfield,[3] Russell M Viner [1], Simon Russell [1]

¹Great Ormond Street Institute of Child Health, University College London, London, UK
²Institute of Epidemiology and Health Care, University College London, London, UK
³Institute of Education, University College London, London, UK

**Correspondence to**
Dr Simon Russell;
s.russell@ucl.ac.uk

## ABSTRACT

**Objectives** Childhood obesity rates in the UK are high. The early years of childhood are critical for establishing healthy behaviours and offer interventional opportunities. We aimed to identify studies evaluating the impact of UK-based obesity interventions in early childhood.

**Design** Systematic review using the Preferred Reporting Items for Systematic Reviews and Meta-Analyses guidelines.

**Data sources** Nine databases were searched in March 2023. Eligibility criteria: We included UK-based obesity intervention studies delivered to children aged 6 months to 5 years that had diet and/or physical activity components and reported anthropometric outcomes. The primary outcome of interest was z-score Body Mass Index (zBMI) change (within and between subjects). Studies evaluating the effects of breastfeeding interventions were not included as obesity prevention interventions, given that best-practice formula feeding is also likely to encourage healthy growth. The publication date for studies was limited to the previous 12 years (2011–23), as earlier reviews found few evaluations of interventions in the UK.

**Data extraction and synthesis** The reviewers worked independently using standardised approach to search, screen and code the included studies. Risk of bias was assessed using Cochrane tools (ROB 2 or ROBINS-I).

**Results** Six trials (five studies) were identified, including two randomised controlled trials (RCT), one cluster randomised trial (CRT), two feasibility CRTs and one impact assessment. The total number of participants was 566. Three trials focused on disadvantaged families and two included high-risk children categorised as having overweight or obesity. Compared with baseline, five interventions reported reductions in zBMI, three of which were statistically significant (p<0.05). Compared with control, five interventions showed zBMI reductions, one of which was significant. Only two trials were followed up beyond 12 months. All studies were found to have a high risk of bias. Meta-analysis was not possible due to the heterogeneity of studies.

**Conclusion** UK evidence was limited but some interventions showed promising results in promoting healthy growth. As part of a programme of policies, interventions in the early years may have an important role in reducing the risk of childhood obesity.

**PROSPERO registration number** CRD42021290676

## STRENGTHS AND LIMITATIONS OF THIS STUDY

⇒ A key strength was that searches were comprehensive and were conducted across nine scientific databases.

⇒ For this review, we used specialist software (EPPI Reviewer Web), which was both a strength and limitation. The software used 'active learning' to enable priority screening, which greatly reduced screening time but resulted in a proportion of studies being excluded without being screened.

⇒ A limitation of our approach was that we excluded studies published prior to 2011; however, previous research suggests little to no evaluations of UK obesity interventions in the early years prior to this time.

⇒ A further limitation was that evaluations of interventions delivered outside the UK were excluded; however, the work has high relevance to UK contexts, particularly in informing policy and practice.

⇒ A final limitation was that we were unable to perform meta-analysis due to the heterogeneity of the interventions.

## INTRODUCTION

In England, data from the National Child Measurement Programme (NCMP) show that the prevalence of obesity among children aged 4–5 years, though plateaued, remains stably high. Disadvantaged children are the worst affected, as the inequalities in childhood obesity are increasing.[1 2] Once established, obesity tracks strongly through childhood, adolescence and into adulthood where it is associated with increased risk of physical and mental health morbidities.[3–7] Prospective evidence indicates that half of the adolescents with obesity had being living with overweight or obesity since the age of 5.[8] Data from the NCMP suggest that the odds of children with overweight in the age of 4–5 years developing obesity by 10 and 11 years were 13 times higher compared with children with healthy BMI.[9] Longitudinal evidence shows that a child's weight status remains stable through primary school years, as 68%

of children with obesity and 78% of children with healthy weight at 4 and 5 years maintained their weight status by 10 and 11 years.[9] The early years of life are therefore a critical period for the development of a healthy weight, representing a window of opportunity for the introduction of a healthy lifestyle to support healthy growth.

Diet[10 11] and physical activity[12] are the major determinants of weight in young children. The diets of many preschoolers in the UK do not adhere to government dietary guidelines, exceeding intake of free sugar and saturated fat, and not reaching recommendations for fruit and vegetables.[13 14] Physical activity levels are reported to be below recommended guidelines[15] with most children not engaging in the 180 min of physical activity per day recommended by the National Health Service (NHS).[16] Lifestyle behaviours track through childhood, into adolescence and adulthood, meaning that those children are likely to continue following unhealthy practices, thereby being at even greater risk of overweight and obesity.[17 18]

Considering the significance of the first 5 years for outcomes later in life, interventions to support the establishment of a healthy weight and a healthy lifestyle are valuable. In the UK, the National Institute for Health and Care Excellence (NICE)[19] gives guidance on the key interventional components to reduce the risk or manage overweight and obesity. Principally, these include the adoption of a healthy diet, addressing lifestyle risk factors within the family and social settings, ie preschools, reducing sedentary lifestyle and supporting behavioural change. The UK Government's Tackling Obesity Policy Paper[20] aims to place prevention at the heart of the health agenda to proactively tackle the burden of obesity for adults and children. The paper suggests that an integrated, whole systems approach is likely to be the most effective in addressing the complex pathways to obesity. It is acknowledged that controlling the problem of childhood obesity is a shared responsibility of the government, local authorities and the NHS, in addition to the food industry, schools and local communities.[20] Preventative and treatment interventions for obesity in early childhood can occur in various settings, typically family homes, nurseries, preschool or healthcare settings. Interventions in community settings, such as nurseries or preschools, are likely to be preventative and universal (for all children), and have the potential to reach more children.[21 22]

The high prevalence of childhood obesity in the UK supports the need for early interventions. To inform such interventions, it is crucial to understand which interventions have been successful. It is also important to identify interventions with longer follow-up periods. The aim of this review was to synthesise evidence of interventions implemented and evaluated in the UK that aim to prevent or treat obesity in children aged 6 months to 5 years. We restricted this review to UK interventions as early childhood environments can differ markedly across countries. Patient and public involvement workshops were conducted to help inform our approach and interpret findings.

## MATERIALS AND METHODS

### Design

This systematic review adhered to the Preferred Reporting Items for Systematic Reviews and Meta-Analyses statement[23] and presents the UK findings of the PROSPERO-registered protocol CRD42021290676.

### Search strategy

In November 2021, we searched nine databases for obesity interventions among children under 5 years old published in peer-reviewed journals (selected databases and search terms used are shown in online supplemental appendix 1). The searches were updated in March 2023. In addition to database searches, the authors searched reference lists of similar reviews. No language restrictions were applied. Our searches included international literature; however, here, we report findings from UK studies only. Applicability and transferability of the interventions conducted elsewhere might be limited due to variability in obesity prevalence, resource availability, policy landscape and delivery of the programmes.[24]

### Inclusion and exclusion criteria

Primary research studies were included if they fulfilled the following criteria:

Population: children aged 6 months–5 years.

Intervention: Pre-intervention and post-intervention studies including randomised controlled trials, quasi-experimental studies, (individual, family- or community-based).

Comparison: pre-intervention vs post-intervention and/or intervention vs control group.

Outcomes: at least one anthropometric outcome (eg, BMI, BMI z-score, BMI percentile, waist circumference, skinfold measurements). Anthropometric measurements were chosen as an outcome because these are the most robust way of assessing impact, compared with other potential outcomes (eg, diet or physical activity).

The search was limited to studies published during the previous 12 years (2011–23) to reflect current or recent interventions and programmes, to understand what works in the current context. Previous reviews found no studies conducted in the UK, meaning it is unlikely that there were UK interventions robustly evaluated prior to 2011.[25 26]

### Selection of studies and data extraction

Search results were imported into Endnote, where duplicates were removed, before being imported into EPPI Reviewer Web,[27] in which further duplicate searches were undertaken. Two reviewers (SM and MS) independently double screened on title and abstract. Using EPPI-Reviewer software, we applied an 'active learning approach', where the prioritisation of records was periodically refreshed during screening so that the most relevant articles were screened first.[28] A graphical output was used to indicate when to stop screening, ie when the number of relevant studies had plateaued (online supplemental

appendix 2). A classifier model was then built using the machine learning algorithm and applied to unscreened items, which generates a score (0–100) indicating relevance; this process reduces the likelihood that relevant studies would be missed. Reviewers independently screened all records with a score higher than 30.

Searches included children aged 0–5 years; interventions that targeted infant feeding in the first 6 months of life were excluded. Evidence suggests that breastfeeding is optimal in terms of growth, development, maturation of the immune system and programming of the metabolic system and that formula feeding can be associated with rapid weight gain and later risk of obesity.[29] However, the association is complex and there is research that suggests best-practice formula feeding can reduce rapid weight gain.[30] For this reason and owing to the complexity of the evidence, we did not include interventions that promote breastfeeding as obesity prevention interventions. The aim of this work was to assess the evidence of interventions promoting health growth in the early years for both breast and formula-fed children. Reviewers developed a data extraction tool and data were doubly extracted. The data extracted included information on trial identification (name of study, authors, year of publication), country (country the trial took place and country of research), trial description, trial duration and follow-up, trial characteristics (preventative/treatment, intervention/programme), participants' characteristics (numbers, ages) and results regarding changes on zBMI. The reviewers double screened independently, and any disagreements were resolved through discussion.

## Data synthesis

We performed a narrative synthesis of findings, summarising the effect estimates; meta-analysis was not possible owing to heterogeneity in intervention approaches, that is, components of the interventions, participants targeted, site of delivery, duration, intensity and length of follow-up. The included studies are presented in two sections: 'preventative' or 'treatment', based on their inclusion criteria. Universal interventions with children of any weight status are included in the 'preventative' category, while interventions that recruited children with overweight or obesity only are classified as 'treatment' interventions. We report the mean difference of any anthropometric outcomes both between-group (ie, intervention vs control) and within-group (ie, before vs after). Measurements obtained at any time point (eg, mid-intervention, right after the intervention, after the end of the intervention) were considered for this review. If studies provided the mean value of anthropometric outcomes before and after the intervention or in the control and the intervention group, two reviewers (SM and MS) calculated the mean difference and the SD of the change using formulas that are suggested by the Cochrane Handbook.[31] The alpha level of significance was $p < 0.05$.

## Risk of bias assessment

Two reviewers (SM and MS) independently assessed the bias using the Cochrane risk of bias assessment tool (RoB 2) to evaluate the included randomised trials.[32] For one study that was an impact assessment, the ROBINS-I tool was used which is suitable for non-randomised studies.[33] Detailed bias assessment can be found in online supplemental appendix 3.

## Patient and public involvement

We conducted meaningful patient and public engagement to integrate the voice of parents throughout this study. Through the National's Children's Bureau, we recruited parents with relevant lived experience from the Family Research Advisory Group panel.[30] Parents from this panel have received training and understand research approaches and have experience of contributing insights and advice to research projects. Six members of the public (five female and one male) agreed to contribute to this study, all were or had been parents of young children, two were also grandparents. Public contributors were diverse in terms of age and were recruited from across England. We held two online sessions with the same parents, one at the outset of the project, to incorporate views into our general approach, and one at the end, to share and discuss interpretations of the research. Specifically, we discussed current practices and evidence of obesity interventions for young children in the UK, what parents perceived to be the effective elements or components of interventions, the challenges of implementing effective interventions and for hard-to-reach groups, how to effectively evaluate interventions in the early years, and the role of parents in supporting interventions that promote healthy growth in early childhood.

# RESULTS
## Description of the studies

In total, the literature search yielded 34 572 potentially relevant articles. After duplicates removal, 21 547 articles were eligible for screening. Two independent reviewers (SM and MS) manually screened 9538 records, in title and abstract, of which 6217 were manually removed (figure 1). Any differences were resolved by reconciliation and, if necessary, in consultations with the other authors. An additional 12 004 records were removed based on title and abstract, as indicated by the machine learning algorithm. A total of 3326 records were included for full-text screening, two of which could not be retrieved, as we could not gain access to them through the University of London Library. Out of the 3324 remaining records, 3108 were manually excluded (reasons for exclusion presented in figure 1), leaving 217 relevant studies. Out of the 217 international studies, five studies (six trials) were conducted in the UK and were included in the final review.[34–38]

One study was conducted in Lancashire,[34] one in Glasgow,[35] one in two areas in North and Central

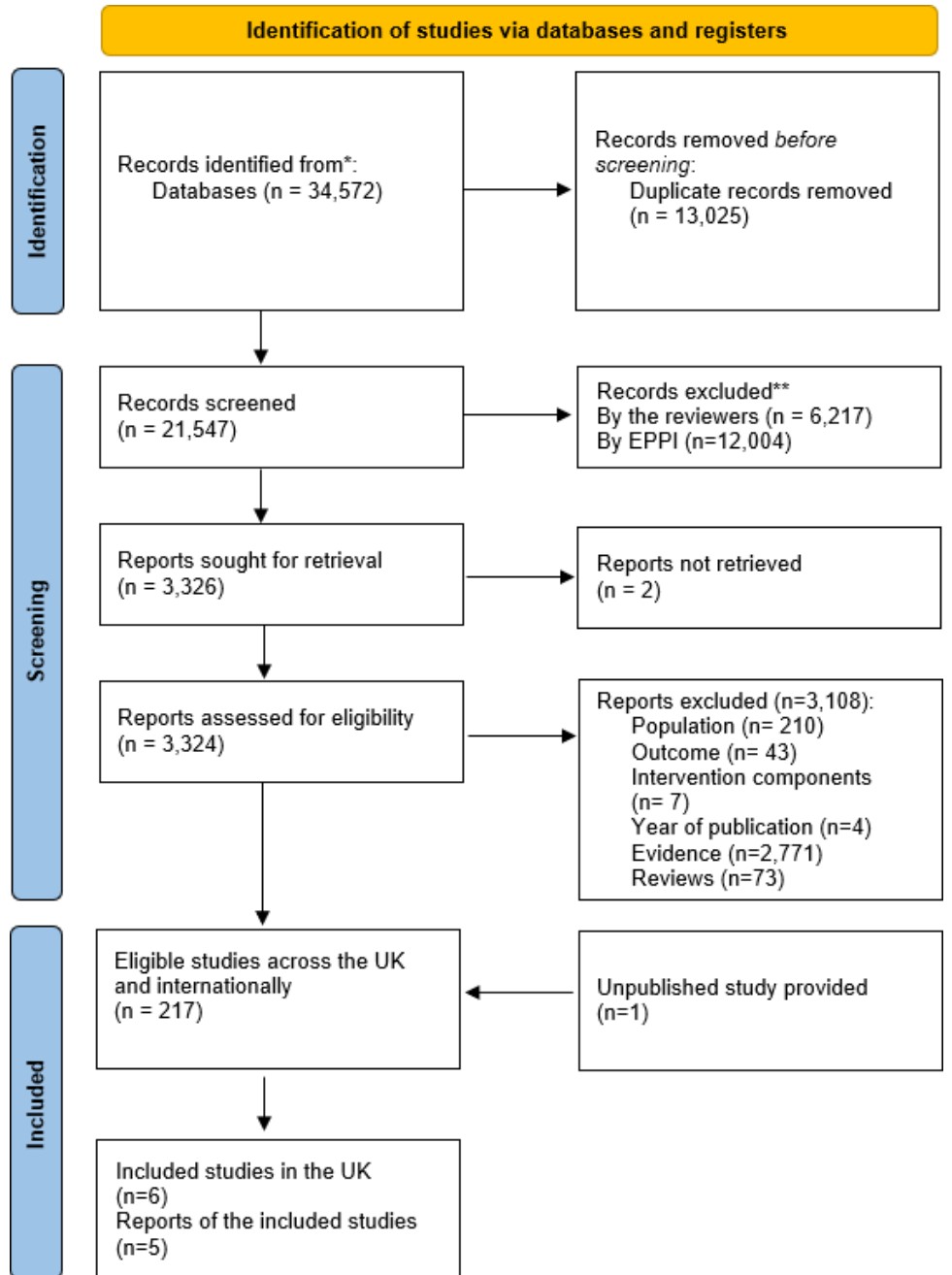

**Figure 1** Flowchart of the included studies.

England,[36] one in Cornwall[37] and one in Hertfordshire.[38] There were two randomised controlled trials,[38] one cluster randomised trial (CRT),[34] two feasibility CRTs[35 36] and one before/after impact assessment of a weight management programme.[37] All studies described interventions with both diet and physical activity outcomes. Sample sizes ranged from 42 to 117. Intervention duration ranged from 8 weeks to 12 months. The follow-up measurements were obtained during the intervention for one study[35] and after the intervention for the rest, ranging from 6 to 30 months from the start of the intervention. Two trials targeted children living with overweight, obesity or at high risk,[37 38] while the remaining four were preventative interventions,[34–36 38] and targeted children of any weight

status (children in the general population, not necessarily living with overweight or obesity). Detailed information for each of the included trials can be found in table 1.

### Intervention characteristics and effects
#### Preventative interventions
The four preventative trials include the *Be active, Eat healthy/Healthy Heroes*[34] intervention, the *ToyBox*[35] intervention, the *Health Exercise Nutrition for the Really Young (HENRY)*[36] programme, and the *Planet Munch* programme, trial 2.[38]

Three out of four trials (Be Active, Eat healthy/Healthy Heroes, ToyBox, HENRY) focused on disadvantaged families or deprived areas.[34–36] Most[34–36] (Be Active, Eat

**Table 1** Characteristics of the included studies

| Author, year, study type | Participants eligible/targeted | Participants randomised (intervention, control)* | Intervention duration | Time of the follow-up measurements | Participants with valid data in the measurements | Change in mean zBMI first or second measurement vs baseline (SD/CI 95%) | Change in mean zBMI vs control (SD/CI 95%) | When evaluated |
|---|---|---|---|---|---|---|---|---|
| Hodgkinson et al., (2019), Pragmatic CRT[34] | 2–4 years, Mean age=NR | 87 (48, 39) | 6 months | 2 years post baseline | 81 (47, 34) | −0.18† (SD 0.99) | −0.74† (−1.10 to −0.38) | NR (Trial published in 2019) |
| Malden et al, (2019), Feasibility CRT | 3–5 years, Mean age=4.4 | 42 (26, 16) | 18 weeks | 15th–17th week of the intervention | 36 (22, 14) | +0.02 (−0.11 to 0.15) | −0.04 (SD 0.13) | 2018 |
| Bryant et al, (2021), Feasibility CRT | 6 months–5 years, Mean age=NR | 117 (47, 70) | 8 weeks | 11 months post baseline | 99 (39, 60) | −0.9 (SD 0.46) | −1.4 (SD 0.08) | NR (Trial published in 2021) |
| Lanigan et al, (Unpublished) RCT-trial 1 | 1–5 years, with OW/OB/high risk Mean age=2.4 | 88 (49, 39) | 6 months | 6 months post baseline | 64 (36, 28) | −0.2 (−0.3 to −0.03) | −0.2 (−0.5 to 0.05) | (Trial not yet published) |
| | | | | 24 months after the end of the intervention | 42 | −0.3 † (−0.5 to to 0.1), | NR | (Trial not yet published) |
| Lanigan et al, (Unpublished) RCT-trial 2 | 1–5 years, of any weight status. Mean age=2.1 | 85 (38, 47) | 6 months | 6 months post baseline | 64 (27, 37) | −0.02 (−0.2 to 0.2), | −0.09 (−0.4 to 0.2) | (Trial not yet published) |
| | | | | 12 months after the end of the intervention | 39 | −0.3 (−0.7, 0.1), | NR | (Trial not yet published) |
| Tee et al, (2021), Impact assessment | Children aged ≤6 years with severe OB Mean age=4.1 | 147 | 12 months | 12 months post baseline | 41 | −0.5† | NR | 2012–17 |

* Numbers at randomisation.
†significant change in zBMI.
CRT, cluster randomised trial; NA, not applicable; NR, not reported; OB, obesity; OW, overweight; RCT, randomised controlled trial.

healthy/Healthy Heroes, ToyBox, HENRY) were administered by preschool staff, which had been appropriately trained, while one (Planet Munch)[38] was delivered by a trained multidisciplinary team including dietitians, nutritionists, community artists and children's centre staff. Two interventions targeted and made changes in preschools' policies; for example, by informing the food policy[34] (Be Active, Eat healthy/Healthy Heroes) or by increasing the time allocated to physical activity (ToyBox).[35] Interventions also targeted the home environment, for example by providing healthy cooking sessions[34 38] (Be Active, Eat healthy/Healthy Heroes, Planet Munch) or sending materials for home-based physical activities (ToyBox, Planet Munch).[35 38] One study[36] (HENRY) was delivered to parents through workshops organised in early years centres, with recommendations provided on nutrition, physical activity and parenting. Planet Munch[38] included workshops for parents and children, that aimed to help them improve their dietary and physical activity practices and adopt a healthier lifestyle.

All studies involved a control group; in the majority of cases, the control group received standard care (Be Active, Eat healthy/Healthy Heroes, ToyBox, HENRY),[34–36] while in one study (Planet Munch),[38] the control group was a delayed intervention group. The risk assessment revealed that *Be active, eat healthy/Healthy Heroes, ToyBox, HENRY and Planet Munch* had a high-risk of bias, with the most studies having high or moderate risk of bias in the randomisation process and the intervention adherence (online supplemental appendix 3).

Only one trial (Be Active, Eat healthy/Healthy Heroes)[34] found the intervention group to have significantly decreased their zBMI after the intervention, compared with the control group [−0.74, (−1.10 to −0.38)]. Two studies[36 38] also found a lower zBMI in the intervention compared with the control group; however, there was either no statistical comparison between them (HENRY),[36] or the difference was not statistically significant (Planet Munch).[38] In one trial[35] (ToyBox), the zBMI had been raised for both groups; nonetheless, this increase was larger for the control group (table 1).

When looking at zBMI changes within the intervention group, three studies showed reductions, ranging from −0.02 (−0.2 to 0.2) to −0.9 (95% CIs not given) (Be Active, Eat healthy/Healthy Heroes, HENRY, Planet Munch).[34 36 38] In one study, this reduction was statistically significant (Be Active, Eat healthy/Healthy Heroes),[34] in one no statistical comparison was reported (HENRY)[36] and one study found the reduction not to be significant (Planet Munch).[38] Finally, one trial (Toy Box)[35] reported zBMI to have increased by 0.02 (−0.11 to 0.15). A second follow-up measurement took place in one of the studies (Planet Munch)[38]; while there was a decrease compared with baseline, this was not significant (table 1).

## Overweight and obesity treatment interventions

We identified two treatment trials. The first was *The Lifestyles, Eating, and Activity for Families (LEAF)*[37] programme,

which was an impact assessment of a childhood obesity service and the treatment trial of *Planet Munch*.[38] *The LEAF programme* targeted children with severe obesity or severe obesity with a relevant comorbidity. The programme consisted of an initial home visit, workshops and follow-up clinic appointments. A community paediatrician, a specialist dietitian and a physical activity advisor were involved for each child. There was no control group in *LEAF*. In terms of within-subject changes, the *LEAF* programme found a significant reduction of zBMI, following programme participation (table 1). The trial was assessed to have a high risk of bias mainly due to missing data.

The treatment trial of *Planet Munch* targeted children with overweight/obesity or who were high-risk. As with the prevention trial, it was delivered by a multidisciplinary team through healthy lifestyle workshops. Planet Munch included a control group (delayed intervention group). At the end of the intervention, after accounting for age and gender, the intervention group had lower mean zBMI score than the control group. For Planet Munch, there was a significant reduction in the zBMI of the intervention group, from baseline to the end of the intervention, as well as from baseline to the 2-year follow-up (table 1).

## Patient and public involvement

Parents, acting as public contributors, were mixed in their views around the potential impact and effectiveness of obesity interventions in early childhood. Half (n=3) felt that interventions in the early years could be effective, while half were sceptical. All parents thought that preventative interventions had the potential to have the greatest impact and the majority (n=5) were in favour of multicomponent interventions that incorporated dietary and physical activity components, giving that the goal in early life is to encourage healthy lifestyles. Parents reported mixed views in relation to intervention components; three parents favoured educational interventions around living healthier lifestyles, two favoured motivational approaches and one favoured addressing environmental factors to enable healthy growth among children. In terms of the interpretation of findings, all parents were surprised at how few interventions with robust evaluation there were in the UK. All parents reported that the cost of living a healthy lifestyle (eg, buying fresh food) was an important factor and suggested there should be financial help or subsidies for parents of young children, particularly families with low income. There was consensus among parents that interventions should be properly evaluated and better funded to ensure they are robust; for example, by following participants up over longer time frames. There was also consensus among parents that families, early years practitioners (eg, health visitors) and parents of children living with excess weight should be consulted when developing interventions in early childhood.

## Discussion

The aim of this systematic review was to identify and evaluate interventions and programmes designed to prevent or treat obesity among children aged 6 months–5 years, implemented and evaluated in the UK. Six studies met the inclusion criteria, all of which included a combination of diet and physical activity components. Compared with baseline, three interventions reported a reduction in zBMI immediately after completing the intervention[37 38] and four interventions a while after the intervention end,[34 36 38] with the effect ranging from −0.02 to −0.9 for the preventative interventions and from −0.3 to −0.5 for the treatment ones. Three of these reductions were significant[34 37 38] (two of which targeted children with overweight/obesity).[37 38] Compared with the control conditions, five interventions showed a reduction in zBMI,[34–36 38] of which one was significant.[34]

This is the first systematic review that focuses on childhood obesity interventions among children aged 6 months–5 years evaluated in the UK. Our findings support existing evidence from international reviews that multicomponent interventions, with both dietary behaviours and physical activity elements, are effective in reducing zBMI among preschoolers.[26 39 40] Small effects were found at individual level; however, if the interventions are delivered at population level, the effects can be larger and meaningful.[41] The present work contributes to the research by synthesising the evidence of recent interventions and assessing their effects on children of any weight status or on children with overweight and obesity.

The current review highlights a lack of UK interventions with robust evaluation. Only Planet Munch was evaluated through a randomised controlled trial. ToyBox and HENRY were evaluated with feasibility cluster randomised trials without a follow-on full trial being published, and these feasibility trials were not adequately powered to detect zBMI changes. Methodological limitations across the studies identified, such as unclear adherence to the intervention, limit our ability to draw firm conclusions about their impact. Evaluating interventions is complex; it is important not only to ascertain impact, but also when, how and in which conditions and for whom interventions work. A theoretical framework, a process to identify failures in implementation, an assessment of the sample and a range of outcomes, in addition to adaptation to local settings, are important contributory factors to the effectiveness and applicability of an intervention.[42 43]

Of the studies identified in this review, only two were followed up beyond 12 months[34 38] meaning there is very little evidence for longer-term effects. All the studies included in this review were small-scale evaluations and not easily generalisable. While effect sizes give some measure of clinical importance (if adequately powered), statistical significance is affected by the sample size and so has limited importance.[44] A reduction in zBMI may be clinically relevant when setting a child on a healthier growth trajectory and improving health outcomes.[45]

Overall, we found a lack of evaluations of interventions that recorded anthropometric outcomes. There were evaluations that reported indirect indicators of intervention impact including reductions in calorie intake or increases in physical activity; however, these indicators can be prone to bias. Children and parents often underreport their energy intake,[46 47] with greater underreporting often observed among children living with obesity.[48] Subjective measures of physical activity also often have low validity and may be biased due to poor recall or social desirability.[49] There are evaluations of interventions (eg, the HENRY programme) that show effectiveness in changing energy balance behaviours[50] but have not been included in this review.

Within a whole-systems approach, local authorities can play a leading role in influencing health behaviour. However, it is important to recognise that financial constraints can limit local action to predominantly statutory services. Owing to these and other limitations, many local authorities have developed interventions relevant to the early years which have not been subject to robust assessment. These are often brief interventions (eg, cooking classes or baby yoga), for which there is no evidence of meaningful and sustained lifestyle changes.

Published studies also had a high risk of bias, limiting the confidence in their results. Additionally, findings were difficult to compare owing to heterogeneity in study designs, intervention approaches and outcome reporting. Some of the reported reductions in zBMI should be interpreted with caution. For example, children that received the *HENRY* intervention started the programme with higher mean zBMI than those in the control group, which may have influenced the effect size. This highlights the need for appropriately designed RCTs with stated primary outcomes, in order to evaluate obesity interventions. Indicators of eating behaviour, physical activity and other energy balance behaviours are important, anthropometric measures remain the most robust and reliable measure of intervention effectiveness.

Findings from this review suggest that interventions that promote healthy growth in the early years could be effective in reducing the risk of obesity as part of a programme of policies and interventions for young children and families. However, all studies would benefit from evaluation using large-scale RCTs before firm conclusions can be drawn. Unpublished data were provided for two trials relating to the *Planet Munch* (formerly Trim Tots) intervention, which found reductions in zBMI for children living with overweight or obesity or who were at high risk. Importantly, reductions were maintained at longer-term follow-up, up to 24 months after completing the intervention. *Planet Munch* was the only preschool obesity intervention in the UK that complied with all NICE recommendations that is, provided advice on how to achieve a healthy diet, encouraged physical activity and included behaviour change strategies in the whole family.

The *LEAF* Programme was also effective in reducing zBMI but was only delivered to children with severe obesity

and was not evaluated in an RCT. Our findings are consistent with previous work,[26] which found obesity management interventions to be more effective compared with preventative interventions in young children. This is an expected outcome, considering the different populations in treatment and prevention trials. Treatment interventions aim to slow weight gain and normalise zBMI while preventative interventions aim to maintain a healthy growth trajectory. Both aim to promote healthy lifestyle behaviours.

This systematic review has limitations. The initial search strategy included international evidence without applying limits on geographical location or language and thus yielded a high number of records. We used EPPI-Reviewer software to apply an active learning approach, which greatly reduced screening time but resulted in 12 004 records being excluded without being screened. Though we also conducted a hand search, there may be relevant studies that have not been included in this review. We found but did not include a child-care self-assessment intervention that aimed to improved physical activity, oral health and nutrition for children aged 2–4 years and was delivered in nurseries.[51] The study was published but not formally peer reviewed, meaning it did not fulfil our criteria for inclusion. We also excluded studies published before 2011; however, previous systematic reviews have not included any UK studies conducted prior to that point, so it is unlikely that any robustly evaluated interventions had been conducted.

Only a small number of preschool obesity interventions have been implemented and evaluated in the UK and a smaller number have been robustly evaluated in randomised controlled trials. Meanwhile, obesity prevalence and the health inequalities remain high in the UK. Obesity policies and interventions are not always implemented or delivered at scale, and implemented interventions are rarely evaluated robustly. Interventions often target individual behaviour changes without addressing the structural and systemic determinants of obesity.[52] The evaluation outcomes of some interventions are encouraging, but more evidence is needed via larger-scale trials. However, scaling-up interventions can be challenging, as adaptations are often required, to meet real-world contexts. Intervention effects can also decrease when applied to the general population, so adaptations should consider the context, implementation and setting of the intervention.[50 53] More research is required to understand what support might be most useful to UK preschoolers and their families.

## CONCLUSIONS

This review found that there are very few evaluations of childhood obesity interventions in preschool children in the UK. Some interventions reported effect sizes that could be clinically significant if replicated in larger sample sizes. Effect sizes were smaller in preventative interventions where the potential for zBMI change is smaller compared with children living with excess weight at the outset. However, improvements in lifestyle behaviours were reported that may reduce the risk of excess weight gain in childhood. More interventions that aim to prevent and treat obesity in preschool children are needed in the UK. Robust evaluation, ideally in RCTs, will be crucial in order to guide policy makers and inform evidence-based practice in families, in communities and local authorities in the UK.

**Acknowledgements** We thank Dr. Julie Lanigan and all the Planet Munch team that agreed to share with us findings from their unpublished work, entitled: 'The TrimTots Healthy Lifestyle Programme for prevention of obesity in preschool children: evidence from 2 randomised controlled trials'. Also, we would like to thank Dr. Steven Hope for commenting on the final draft of the manuscript. Finally, we would like to thank the Family Research Advisory Group for their contribution in our research.

**Contributors** SR and RV conceived and designed the project. SM, SR and CS designed the search strategy, and SM, MS and CS conducted the searches. SM, MS and JP conducted screening, data extraction and assessment of bias. JL provided data from one unpublished study (two trials). SM, MS and SR analysed the data. All authors contributed to the interpretation of findings. SM drafted the paper and all authors revised and edited the manuscript, before approving the final draft. SR acted as guarantor for the study.

**Funding** This study was funded by the UK National Institute for Health and Care Research Policy Research Programme (PR-PRU-0916-21001; grant number 174868). The views expressed in this Article are those of the authors and not necessarily those of the National Institute for Health and Care Research or the UK Department of Health and Social Care.

**Competing interests** None declared.

**Patient and public involvement** Patients and/or the public were involved in the design, or conduct, or reporting, or dissemination plans of this research. Refer to the Methods section for further details.

**Patient consent for publication** Not applicable.

**Ethics approval** This study was exempt from the requirement of approval by the UCL Ethics Committee as it used secondary data available in the public domain and it is not possible to identify individuals from the information provided.

**Provenance and peer review** Not commissioned; externally peer reviewed.

**Data availability statement** All data relevant to the study are included in the article or uploaded as supplementary information.

**ORCID iDs**
Semina Michalopoulou http://orcid.org/0000-0003-3198-0411
Julie Lanigan http://orcid.org/0000-0002-8339-9947
Russell M Viner http://orcid.org/0000-0003-3047-2247
Simon Russell http://orcid.org/0000-0001-9447-1169

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
