## [Reviewer comments · BMJ Open]

ARTICLE DETAILS

TITLE (PROVISIONAL)	Assessing the impact of obesity interventions in the early years: A systematic review of UK-based studies
AUTHORS	Michalopoulou, Semina; Sifaki, Maria; Packer, Jessica; Lanigan, Julie; Stansfield, Claire; Viner, Russell; Russell, Simon

VERSION 1 – REVIEW

REVIEWER	Bolton, Kristy Deakin University, Institute for Physical Activity and Nutrition
REVIEW RETURNED	25-Sep-2023

GENERAL COMMENTS	Overall this study has potential but a lack of consistent aim, reference to child age group, and terms such as sustainability, feasibility, effectiveness etc make this paper a little hard to follow. The section on public involvement appears without warning/justification in the methods, and hence is not mentioned again in the results or discussion. The risk of bias is also not reported in detail in the methods nor future implications for research in the discussion. I would encourage the authors to think about refining their aim, and being consistent with it and terminology throughout the paper. I would also be cautious about overstating the importance of study findings. Abstract: Line 19: Aim: “We aimed to assess the effectiveness of UK-based childhood obesity interventions.” I am wondering if this should be tweaked slightly to reflect the age range – your title mentions “early years” and the methods mention 0-5 years, so I wonder if the aim would be clearer to the reader if you include “early childhood obesity interventions” to indicate the lifespan of focus? Line 24: “Studies evaluating the effects of breastfeeding or complementary feeding interventions were excluded, as these aim to support healthy growth development”. I query the justification provided as there are many studies that demonstrate association between feeding type (i.e. formula feeding) and rapid weight gain and risk of childhood obesity. So in this instance, whilst strategies focusing on infant feeding may focus on healthy growth/development, they are also childhood obesity prevention interventions? I also then query how the age range of 0-5 years is applicable – perhaps it should be 1-5 years if most studies in 0-1 year old range I expect according to your justification would be excluded? Was having a control arm a part of inclusion criteria?
--

	Line 28: “Bias was assessed using Cochrane tools” – I may be incorrect but these tools assess risk of bias – so I think this should be included in text. Results line 32: “The outcome of interest was zBMI change (within and between subjects).” This appears to be in the wrong section – should this be in the methods section? Results line 34: “three of which were statistically significant” – at what alpha level was the significance determined? Line 36: RE: “Only two trials were followed up beyond 12 months and only one complied with the National Institute’s Health and Care Excellence guidelines”. What is the relevance of the National Institute’s Health and Care Excellence? This appears out of the blue. How was this assessed? Detail should be added to the methods briefly to reflect this, and to the introduction so the reader can follow this more clearly. Abstract does not contain any brief results of the risk of bias – were there domains that were done commonly well/not well? How were the papers assessed overall? High/low risk/some concerns? My query relates to the title of the paper - if assessing the effectiveness of UK-based early childhood obesity interventions, was there a meta-analysis conducted? How are authors assessing effectiveness? Otherwise the title may need refining to reflect the paper more accurately? Strengths and limitations line 43 – global audience may not understand what reception age children are. I would consider examining how children/their age group is referred to in the paper throughout and decide on a consistent term throughout. Strengths and limitations – line 49 “A meta-analysis was not possible due to the heterogeneity of the interventions” – this might be important to include in the body of the abstract. Introduction  • Different age groups are mentioned with regards to children – reception age 4-5 yo, increased weight gain 2-6yo. I would consider tightening your age range of 0-5 to reflect the narrative more clearly. • Line 64 is missing a reference • Line 73 – is the word dietary missing? I.e. government dietary guidelines? “The diets of many pre-schoolers in the UK do not adhere to government dietary guidelines” Otherwise, what guidelines are you referring to? • Line 81 – preschool period – another way to discuss a lifespan/age range. Again, I suggest choosing consistent terminology. • Line 89 is missing a reference • Line 95: “It is also important to identify interventions with sustainable effects, i.e., have a long-term follow-up.” Perhaps the aim should be tweaked to examine the impact AND sustainability of early childhood obesity interventions in the UK? So it’s clear in the aim that follow up is being considered? Was it an inclusion criteria? This was missing from the abstract methods. • Line 97: “early years of life” is vague as discussed above.
--	--

	 • Line 99: what is meant by “UK have prima facie evidence of feasibility”. Is feasibility also being assessed? If so, this needs to be reflected in the study aims too. Methods  • Line 139: “While searches included children aged 0-5, interventions that targeted infant feeding were excluded, as their scope was predominantly infant development rather than obesity prevention/treatment”. As discussed above, I disagree with this – infant feeding has been shown to impact rapid weight gain and risk of overweight/obesity. Perhaps your age range should be 2-5 years old, or 1-5 years old? Not 0-5 years old? How many papers were found on children under 2 years of age? • Lines 133-138 – I am not familiar with this approach. What is the advantage of this active learning approach? I don’t quite understand the purpose apologies. Did the screeners not do all of their screening on their own, but instead used software for some of it? “Using EPPI- 133 Reviewer software, we applied an ‘active learning approach’, where the prioritisation of records 134 was periodically refreshed during screening so that the most relevant articles were screened 135 first.(26) A graphical output was used to indicate when to stop screening, i.e., when the number 136 of relevant studies had plateaued (Appendix 2). A classifier model was then built using the 137 machine learning algorithm and applied to unscreened items, which generates a score 138 indicating relevance; this process reduces the likelihood that relevant studies would be missed”  • Line 149: “We performed a narrative synthesis of findings, summarising the effect estimates; meta-analysis was not possible owing to heterogeneity in intervention approaches, i.e., components of the interventions, participants targeted, site of delivery, duration, intensity, and length of follow-up.” What about feasibility as mentioned in the introduction? • Lines 168-176: I value the incorporation of end users/consumers/lived experience – but how does this relate to the studies that were included in the final pool? Were the parent/public contributors directly involved in the included studies? This part of the study appears without any warning – there was nothing about it in the aim nor introduction. A justification as to how this fits with the study, and study aims reflecting this work, needs to be incorporated. I cannot see the link since the systematic review was examining impact/sustainability, not barriers/enablers to implementation? There almost seems to be two separate pieces of work? Were barriers/enablers to implementation/evaluation extracted out in the data extraction stage to compare/contrast to those public/parent contributors? The purpose/link between these two pieces of work isn’t clear to me as a reader. Patient and Public Involvement 169 “An initial patient and public involvement (PPI) session was conducted at the outset of the 170 project. We discussed with a Family Research Advisory Group, facilitated by the National’s 171 Children’s Bureau (NCB),(30) the evidence of current childhood obesity interventions in the UK
--	---

172 and what parents/public contributors perceived to be the elements or components of
173 interventions that made them effective. A second session was conducted at the end of the
174 project, where review findings were discussed. Parents/public contributors discussed the
175 challenges of implementing effective interventions, how to effectively evaluate interventions in
176 the early years, and the importance of parental involvement in effective interventions

Furthermore, the methods are not detailed enough to allow this to be repeated. How were patients/parents/public recruited? Who is the Family Research Advisory Group, how were they formed? How were they recruited to this study? Do parents/public contributors define "effective" similar to the criteria in this study? In my experience, those on the ground have different measures of success/effect. How was elements/components/effectiveness discussed? Were the meetings recorded? How long were each of the sessions? What was the feedback on the project findings? Was a thematic analysis conducted on the feedback? What does all of this mean? Many many questions for this part of the study which is too brief in purpose and details.

- Methods is missing alpha level of significance.
- Results: line 182-184 "Any differences were resolved by reconciliation and, if necessary, in consultations
183 with the other authors. An additional 12,004 records were removed based on title and abstract,
184 as indicated by the machine learning algorithm." This explanation is what I was looking for above in the methods, this is a much clearer explanation. Was there any cross-checking with the machine learning? i.e. did the machine learning test a proportion of the manual pool extracted, and was there manual checking of a proportion of the machine learning pool?
- Line 185 .."two of which could not be retrieved, as we could not gain access to them" – why couldn't you gain access? Did you try via library or contacting the study authors?
- Line 187-201 – in the summary of the included studies, there is not a description of how many were diet, PA only, or diet + PA interventions?
- The results section should have more detail on why studies were considered to have a high risk of bias – were there common domains that made the studies score highly? There is no detail to understand the domains/elements of the study that had risk of bias. This could also be presented in a table/figure, but needs some more detail in text.
- There are no results relating to the public/parent involvement.

Discussion

- Line 260-262 – the aim is written differently again here – suggest reviewing the aim and writing it consistently throughout the paper, with added elements of sustainability and consideration of using the term "impact of" rather than "effectiveness" given a meta analysis could not be conducted
- Line 275: "The present work contributes to the research and policy by synthesizing the evidence of recent interventions and assessing their effects on children of any weight status or on children with overweight and obesity". What do the authors mean by policy here? If there are very little studies, and not enough to do a meta-analysis, and not enough that have followed up for a

	significant duration, then how can this inform policy? And what type of policy? I would like to understand more what the co-authors are discussion here if they can explain so to the reader?  • Line 320: “Findings from this review do suggest that UK weight management and prevention interventions for the early years can be feasible and effective in reducing the risk of obesity”. I think this statement is an overstatement of the study findings – there are a lack of studies as authors of noted, lack of follow up. How is feasibility measured to provide evidence in this study? And there were mixed findings from the studies so I don’t believe overall they can be considered effective? Perhaps some studies might be promising, but I don’t think they are completely effective, also considering high risk of bias that most studies appear to be assessed with? • Line 347: “The outcomes of some interventions suggest that they are feasible, acceptable, and have the potential to be trialled at a larger scale”. I don’t believe the authors have provided any evidence of interventions being acceptable? And trialled at a larger scale? These appear to be overstatements perhaps? • There is no discussion relating to the public/parent involvement.
--	---

REVIEWER	Dobell, Alexandra University of Birmingham, Applied Health Research
REVIEW RETURNED	26-Oct-2023

GENERAL COMMENTS	I would like to congratulate the authors on a well written and concise systematic review. Although the evidence to answer the question in this review is limited, it has been synthesised well with a narrative approach. It clearly highlights the areas for research and policy development within the UK intervention, research, and policy landscape to address the growing issue of childhood obesity. I have offered some thoughts and possible improvements to work below, set out by each section of the manuscript. Abstract: A few adjustments to make: Line 42- change reception to under 5s or preschoolers. Line 47: remove the word ‘with’ after conducted. Introduction: The introduction is well written and concisely introduces the area, centering on childhood obesity and the need for early childhood intervention. Background to government policy is also given which strengthens this section. Perhaps worth clarifying what age range pre-schoolers includes, and citing previously literature to support this. Methods: A well written section which clearly describes the methods and materials used within the review. The use of machine learning in this method is novel to myself but I think offers a great tool for future studies. PPI section feels redundant, it does not really tell the reader why this was important to this particular piece of work, making it more relevant would be useful. Results: The split of preventative and treatment interventions is useful and highlights the differences in outcomes for these two approaches. However, looking at countrywide intervention and whole-system approaches, preventative measures are more likely to be implemented in preschool populations, particularly thinking about childcare approaches in the UK.
--

	Discussion: Line 271: you mention 'reception-aged' children, this need to be changed to reflect the 0-5 years age range. Perhaps 'This is the first systematic review that focuses on childhood obesity interventions among children aged 0-5 years evaluated in the UK' Also consider the use of preschooler, as some would not consider the infant age range 0-1 as a preschooler- make sure any changes to terms used are consistent throughout the manuscript. Line 316: you mention that the children in the control group gained weigh during the intervention, as children are growing rapidly at this age, is an increase in weight not to be expected? Line 318: are you saying these primary outcomes should be related to BMI and/or anthropometrics rather than PA and calorie measurements? Limitations: you mention that literature may have been missed. One intervention that springs to mind that is not covered is the NAP SACC UK feasibility trial which was based in nurseries https://pubmed.ncbi.nlm.nih.gov/31343857/. This study also used a method of calorie intake that was less prone to bias, as well as objective PA measures- which was discussed in lines 299 onwards. Overall, I think some thoughts around why there is a lack of intervention studies based in the UK should be touched upon within this section to make sure this is really context specific.
--	--

VERSION 1 – AUTHOR RESPONSE

Reviewer: 1 Dr. Kristy Bolton, Deakin University	
Overall this study has potential but a lack of consistent aim, reference to child age group, and terms such as sustainability, feasibility, effectiveness etc make this paper a little hard to follow. The section on public involvement appears without warning/justification in the methods, and hence is not mentioned again in the results or discussion. The risk of bias is also not reported in detail in the methods nor future implications for research in the discussion. I would encourage the authors to think about refining their aim, and being consistent with it and terminology throughout the paper. I would also be cautious about overstating the importance of study findings.	We have made the aim clear and consistent through the document. We have also made edits to ensure child age group (children aged 6 months – 5 years) is consistent throughout. We have replaced the word effectiveness with impact in the title and in the main text. We have now updated the PPI parts and incorporated smoothly into the paper. We have added details regarding the bias assessment, and we have added a reference to Appendix 3, where detailed tables are presented. We have changed the wording to avoid overstating the importance of the findings.
Abstract	
Line 19: Aim: "We aimed to assess the effectiveness of UK-based childhood obesity interventions." I am wondering if this should be tweaked slightly to reflect the age range – your title	We have now specified that 'We aimed to identify studies evaluating the impact of UK-based obesity interventions in early childhood.' (Page2, §1)

mentions “early years” and the methods mention 0-5 years, so I wonder if the aim would be clearer to the reader if you include “early childhood obesity interventions” to indicate the lifespan of focus?	
Line 24: “Studies evaluating the effects of breastfeeding or complementary feeding interventions were excluded, as these aim to support healthy growth development”. I query the justification provided as there are many studies that demonstrate association between feeding type (i.e. formula feeding) and rapid weight gain and risk of childhood obesity. So in this instance, whilst strategies focusing on infant feeding may focus on healthy growth/development, they are also childhood obesity prevention interventions? I also then query how the age range of 0-5 years is applicable – perhaps it should be 1-5 years if most studies in 0-1 year old range I expect according to your justification would be excluded?	We accept the reviewers point here and have added further clarification and justification for our study design. The text now reads: “Studies evaluating the effects of breastfeeding interventions were not included as obesity prevention interventions, given that best practice formula feeding is also likely to encourage healthy growth.” (Page 2, §3) We agree that the age range of 0-5 years is misleading given we have excluded breastfeeding and weaning interventions. We have changed the age range to 6 months-5 years throughout. Given that one programme (HENRY) focused on healthy lifestyle intervention for children aged 6-11 months
Was having a control arm a part of inclusion criteria?	We have now included a sentence that clarifies that we were interested in within and between subject changes. (Page 2, §3)
Line 28: “Bias was assessed using Cochrane tools” – I may be incorrect but these tools assess risk of bias – so I think this should be included in text.	We have corrected the term in the text. (Page 2, §4)
Results line 32: “The outcome of interest was zBMI change (within and between subjects).” This appears to be in the wrong section – should this be in the methods section?	We moved this sentence to the methods. (Page 2, §3)
Results line 34: “three of which were statistically significant” – at what alpha level was the significance determined?	The level of significance was p value < 0.05 – which has now been detailed in the abstract. (Page 2, §5)
Line 36: RE: “Only two trials were followed up beyond 12 months and only one complied with the National Institute’s Health and Care Excellence guidelines”. What is the relevance of the National Institute’s Health and Care Excellence? This appears out of the blue. How was this assessed? Detail should be added to the methods briefly to reflect this, and to the introduction so the reader can follow this more clearly.	The sentence about NICE is now removed from the abstract. We provide details about the NICE Guidance in the main text (i.e. introduction) (Page 5, §2)

Abstract does not contain any brief results of the risk of bias – were there domains that were done commonly well/not well? How were the papers assessed overall? High/low risk/some concerns?	We have added that all studies were found to have high risks of bias. We have provided more details in the main text and in Appendix 3. (Page 3, §1)
My query relates to the title of the paper - if assessing the effectiveness of UK-based early childhood obesity interventions, was there a meta-analysis conducted? How are authors assessing effectiveness? Otherwise the title may need refining to reflect the paper more accurately?	We have now refined the title of the paper and we have replaced effectiveness with impact.
Strengths and limitations line 43 – global audience may not understand what reception age children are. I would consider examining how children/their age group is referred to in the paper throughout and decide on a consistent term throughout.	We have changed 'reception aged children' to 'children aged 6 months to 5 years' for consistency throughout the manuscript.
Strengths and limitations – line 49 “A meta-analysis was not possible due to the heterogeneity of the interventions” – this might be important to include in the body of the abstract.	We have now added this sentence to the abstract. (Page 3, §1)
Introduction	
Different age groups are mentioned with regards to children – reception age 4-5 yo, increased weight gain 2-6yo. I would consider tightening your age range of 0-5 to reflect the narrative more clearly.	We have clarified the text in terms of age groups. We have also updated the age range to between 6 months-5 years.
Line 64 is missing a reference	We have attended to this.
Line 73 – is the word dietary missing? I.e. government dietary guidelines? “The diets of many pre-schoolers in the UK do not adhere to government dietary guidelines” Otherwise, what guidelines are you referring to?	We have clarified that we are describing dietary guidelines. (page4, §2)
Line 81 – preschool period – another way to discuss a lifespan/age range. Again, I suggest choosing consistent terminology.	We have made terminology consistent throughout the paper.
Line 89 is missing a reference	This reference has now been added.
Line 95: “It is also important to identify interventions with sustainable effects, i.e., have a long-term follow-up.” Perhaps the aim should be tweaked to examine the impact AND sustainability of early childhood obesity	We have now edited the language; we discuss follow up length without making inferences about sustainability. We have clarified that longer follow up was not one of the inclusion criteria. Trials with longer follow up periods were identified as they

interventions in the UK? So it's clear in the aim that follow up is being considered? Was it an inclusion criteria? This was missing from the abstract methods.	had the potential to demonstrate whether weight change was maintained. (Page5, §2)
Line 97: "early years of life" is vague as discussed above.	We have now made it consistent throughout the paper that we are referring to children aged 6 months-5 years.
Line 99: what is meant by "UK have prima facie evidence of feasibility". Is feasibility also being assessed? If so, this needs to be reflected in the study aims too.	We have now removed this sentence as feasibility was not one of the study aims. (Page6, §1)
Methods	
Line 139: "While searches included children aged 0-5, interventions that targeted infant feeding were excluded, as their scope was predominantly infant development rather than obesity prevention/treatment". As discussed above, I disagree with this – infant feeding has been shown to impact rapid weight gain and risk of overweight/obesity. Perhaps your age range should be 2-5 years old, or 1-5 years old? Not 0-5 years old? How many papers were found on children under 2 years of age?	We have now added a clearer justification. The text now reads: 'Searches included children aged 0-5 years, interventions that targeted infant feeding in the first 6 months of life were excluded. Evidence suggests that breastfeeding is optimal in terms of growth, development, maturation of the immune system and programming of the metabolic system, and that formula feeding can be associated with rapid weight gain and later risk of obesity.(29) However, the association is complex and there is research that suggests best practice formula feeding can reduce rapid weight gain.(30) For this reason and owing to the complexity of the evidence, we did not include interventions that promote breastfeeding as obesity prevention interventions. The aim of this work was to assess the evidence of interventions promoting health growth in the early years for both breast and formula fed children'. (Pages 8, §1)
Lines 133-138 – I am not familiar with this approach. What is the advantage of this active learning approach? I don't quite understand the purpose apologies. Did the screeners no do all of their screening on their own, but instead used software for some of it? "Using EPPI- 133 Reviewer software, we applied an 'active learning approach', where the prioritisation of records 134 was periodically refreshed during screening so that the most relevant articles were screened 135 first.(26) A graphical output was used to indicate when to stop screening, i.e., when the number	We have now clarified that that: 'A classifier model was then built using the machine learning algorithm and applied to unscreened items, which generates a score (0-100) indicating relevance; this process reduces the likelihood that relevant studies would be missed. Reviewers independently screened all records with a score higher than 30.' (Page 7, §3)

136 of relevant studies had plateaued (Appendix 2). A classifier model was then built using the 137 machine learning algorithm and applied to unscreened items, which generates a score 138 indicating relevance; this process reduces the likelihood that relevant studies would be missed”	
Line 149: “We performed a narrative synthesis of findings, summarising the effect estimates; meta-analysis was not possible owing to heterogeneity in intervention approaches, i.e., components of the interventions, participants targeted, site of delivery, duration, intensity, and length of follow-up.” What about feasibility as mentioned in the introduction?	Feasibility was not an aim of the study, so we have removed mention of feasibility in the introduction.
Lines 168-176: I value the incorporation of end users/consumers/lived experience – but how does this relate to the studies that were included in the final pool? Were the parent/public contributors directly involved in the included studies? This part of the study appears without any warning – there was nothing about it in the aim nor introduction. A justification as to how this fits with the study, and study aims reflecting this work, needs to be incorporated. I cannot see the link since the systematic review was examining impact/sustainability, not barriers/enablers to implementation? There almost seems to be two separate pieces of work? Were barriers/enablers to implementation/evaluation extracted out in the data extraction stage to compare/contrast to those public/parent contributors? The purpose/link between these two pieces of work isn’t clear to me as a reader. Patient and Public Involvement 169 “An initial patient and public involvement (PPI) session was conducted at the outset of the 170 project. We discussed with a Family Research Advisory Group, facilitated by the National’s 171 Children’s Bureau (NCB),(30) the evidence of current childhood obesity interventions in the UK 172 and what parents/public contributors perceived to be the elements or components	We have now updated the PPI part in our paper. We have added a sentence in the introduction about PPI (page 6, §1) , a section under the methods that discusses the patient and public involvement in the study (pages 9-10, §3) and a paragraph in the results about the findings of the PPI workshops (page 14,§1).

of 173 interventions that made them effective. A second session was conducted at the end of the 174 project, where review findings were discussed. Parents/public contributors discussed the 175 challenges of implementing effective interventions, how to effectively evaluate interventions in 176 the early years, and the importance of parental involvement in effective interventions	
Furthermore, the methods are not detailed enough to allow this to be repeated. How were patients/parents/public recruited? Who is the Family Research Advisory Group, how were they formed? How were they recruited to this study? Do parents/public contributors define “effective” similar to the criteria in this study? In my experience, those on the ground have different measures of success/effect. How was elements/components/effectiveness discussed? Were the meetings recorded? How long were each of the sessions? What was the feedback on the project findings? Was a thematic analysis conducted on the feedback? What does all of this mean? Many many questions for this part of the study which is too brief in purpose and details.	We have now updated the PPI part in our paper. We have added a sentence in the introduction about PPI (page 6, §1) , a section under the methods that discusses the patient and public involvement in the study (pages 9-10, §3) and a paragraph in the results about the findings of the PPI workshops (page 14,§1).
Methods is missing alpha level of significance.	We have now added the alpha level of significance. (Page 9, §1)
Results:	
line 182-184 “Any differences were resolved by reconciliation and, if necessary, in consultations 183 with the other authors. An additional 12,004 records were removed based on title and abstract, 184 as indicated by the machine learning algorithm.” This explanation is what I was looking for above in the methods, this is a much clearer explanation. Was there any cross-checking with the machine learning? i.e. did the machine learning test a proportion of the manual pool extracted, and	The machine did not cross-check our screening but used our screening decisions to inform the active learning approach. Reviewers cross checked the machine decisions by screening following the application of the classifier.

was there manual checking of a proportion of the machine learning pool?	
Line 185 .."two of which could not be retrieved, as we could not gain access to them" – why couldn't you gain access? Did you try via library or contacting the study authors?	We attempted to gain access via the University of London Library but were not successful. This detail has been added to the main text. (Page 10, §2)
Line 187-201 – in the summary of the included studies, there is not a description of how many were diet, PA only, or diet + PA interventions?	We have now added a sentence that all of the included interventions had both diet and physical activity components. (Page 10, §3)
The results section should have more detail on why studies were considered to have a high risk of bias – were there common domains that made the studies score highly? There is no detail to understand the domains/elements of the study that had risk of bias. This could also be presented in a table/figure, but needs some more detail in text.	We have provided further detail regarding the areas in which studies were assessed to have a high risk of bias. We have also added a reference to Appendix 3, where tables detailing risk of bias are presented. (Page 12, §2)
There are no results relating to the public/parent involvement.	As described, we have now updated the PPI part in our paper. We have added a sentence in the introduction about PPI (page 6, §1) , a section under the methods that discusses the patient and public involvement in the study (pages 9-10, §3) and a paragraph in the results about the findings of the PPI workshops (page 14,§1).
Discussion	
Line 260-262 – the aim is written differently again here – suggest reviewing the aim and writing it consistently throughout the paper, with added elements of sustainability and consideration of using the term "impact of" rather than "effectiveness" given a meta analysis could not be conducted	We have now stated the aim of the review consistently throughout the paper. We have also replaced the term effectiveness with the term impact in the title and main text. (Page 14, §2)
Line 275: "The present work contributes to the research and policy by synthesizing the evidence of recent interventions and assessing their effects on children of any weight status or on children with overweight and obesity". What do the authors mean by policy here? If there are very little studies, and not enough to do a meta-analysis, and not enough that have followed up for a significant duration, then how can this inform policy? And what type of policy? I would like to understand more what the co-authors are	We have removed reference to 'policy' here. We have discussed policy implication on page 18, §1.

discussion here if they can explain so to the reader?	
Line 320: “Findings from this review do suggest that UK weight management and prevention interventions for the early years can be feasible and effective in reducing the risk of obesity”. I think this statement is an overstatement of the study findings – there are a lack of studies as authors of noted, lack of follow up. How is feasibility measured to provide evidence in this study? And there were mixed findings from the studies so I don't believe overall they can be considered effective? Perhaps some studies might be promising, but I don't think they are completely effective, also considering high risk of bias that most studies appear to be assessed with?	We have edited the sentence to avoid overstating findings. The text now reads: ‘Findings from this review suggest that interventions that promote healthy growth in the early years could be effective in reducing the risk of obesity as part of a programme of policies and interventions for young children and families.’ (Page 17, §2)
Line 347: “The outcomes of some interventions suggest that they are feasible, acceptable, and have the potential to be trialled at a larger scale”. I don't believe the authors have provided any evidence of interventions being acceptable? And trialled at a larger scale? These appear to be overstatements perhaps?	We have now removed the words feasible and acceptable and we clarified that the outcomes were encouraging but more evidence is needed. The text now reads: ‘The evaluation outcomes of some interventions are encouraging, but more evidence is needed via larger scale trials.’ (Page18, §1)
There is no discussion relating to the public/parent involvement.	As described, we have now updated the PPI part in our paper. We have added a sentence in the introduction about PPI (page 6, §1) , a section under the methods that discusses the patient and public involvement in the study (pages 9-10, §3) and a paragraph in the results about the findings of the PPI workshops (page 14,§1).
Reviewer: 2 Dr. Alexandra Dobell, University of Birmingham	
I would like to congratulate the authors on a well written and concise systematic review. Although the evidence to answer the question in this review is limited, it has been synthesised well with a narrative approach. It clearly highlights the areas for research and policy development within the UK intervention, research, and policy landscape to address the growing issue of childhood obesity. I have offered some thoughts and possible improvements to work below, set out by each section of the manuscript.	Thank you. Your feedback has been very helpful for us.
Abstract: A few adjustments to make:	

Line 42- change reception to under 5s or preschoolers.	We have changed the 'reception aged children' to 'children aged 6 months to 5 years'
Line 47: remove the word 'with' after conducted.	The extra word has now been deleted.
Introduction:	
The introduction is well written and concisely introduces the area, centering on childhood obesity and the need for early childhood intervention. Background to government policy is also given which strengthens this section. Perhaps worth clarifying what age range pre-schoolers includes, and citing previously literature to support this.	We now refer to children aged 6months to 5 years consistently throughout the manuscript.
Methods:	
A well written section which clearly describes the methods and materials used within the review. The use of machine learning in this method is novel to myself but I think offers a great tool for future studies. PPI section feels redundant, it does not really tell the reader why this was important to this particular piece of work, making it more relevant would be useful.	As described, we have now updated the PPI part in our paper. We have added a sentence in the introduction about PPI (page 6, §1) , a section under the methods that discusses the patient and public involvement in the study (pages 9-10, §3) and a paragraph in the results about the findings of the PPI workshops (page 14,§1).
Results:	
The split of preventative and treatment interventions is useful and highlights the differences in outcomes for these two approaches. However, looking at countrywide intervention and whole-system approaches, preventative measures are more likely to be implemented in preschool populations, particularly thinking about childcare approaches in the UK.	We have now added two sentences in the introduction to address this comment. The text now reads: 'Preventative and treatment interventions for obesity in early childhood can occur in various settings, typically family homes, nurseries, pre-school, or healthcare settings. Interventions in community settings, such as nurseries or pre-schools, are likely to be preventative and universal (for all children), and have the potential to reach more children.(21,22)' (Page 5, §2)
Discussion	
Line 271: you mention 'reception-aged' children, this need to be changed to reflect the 0-5 years age range. Perhaps 'This is the first systematic review that focuses on childhood obesity interventions among children aged 0-5 years evaluated in the UK' Also consider the use of preschooler, as some would not consider the infant age range 0-1 as a preschooler- make sure any	We have changed 'reception aged' children to 'children aged 6 months -5 years. Reference to age is now consistent throughout the manuscript.

changes to terms used are consistent throughout the manuscript.	
Line 316: you mention that the children in the control group gained weigh during the intervention, as children are growing rapidly at this age, is an increase in weight not to be expected?	We have now removed this sentence.
Line 318: are you saying these primary outcomes should be related to BMI and/or anthropometrics rather than PA and calorie measurements?	We have now added a sentence for clarity. The text reads: 'This highlights the need for appropriately designed RCTs with stated primary outcomes, in order to evaluate obesity interventions. Indicators of eating behaviour, physical activity and other energy balance behaviours are important, anthropometric measures remain the most robust and reliable measure of intervention effectiveness.' (Page 17, §1)
: Limitations: you mention that literature may have been missed. One intervention that springs to mind that is not covered is the NAP SACC UK feasibility trial which was based in nurseries https://pubmed.ncbi.nlm.nih.gov/31343857/ . This study also used a method of calorie intake that was less prone to bias, as well as objective PA measures- which was discussed in lines 299 onwards.	Thank you for mentioning the intervention we have missed. We checked our records and realised that we had identified this study as an abstract only, which was excluded (i.e. we did not include abstracts). The study is published in a NIHR Journal and not to a peer reviewed journal, which was one of our criteria. (Page 18, §3)
Overall, I think some thoughts around why there is a lack of intervention studies based in the UK should be touched upon within this section to make sure this is really context specific.	We have now added a sentence in the limitations section relating to the lack of interventions in the UK. (Page 19, §1)

VERSION 2 – REVIEW

REVIEWER	Dobell, Alexandra University of Birmingham, Applied Health Research
REVIEW RETURNED	15-Dec-2023
GENERAL COMMENTS	Thank you to the authors for making the suggested changes to the manuscript, I believe these add to the quality of the work that has been produced. I would only recommend that the PPI element of this paper is revisited, as also suggested by reviewer 1, it does feel disjointed and there is still a lack of information surrounding the recruitment

	of the group and their demographics, which are important elements related to this research.
--	---

VERSION 2 – AUTHOR RESPONSE

Reviewer: 2 Dr. Alexandra Dobell, University of Birmingham	
I would only recommend that the PPI element of this paper is revisited, as also suggested by reviewer 1, it does feel disjointed and there is still a lack of information surrounding the recruitment of the group and their demographics, which are important elements related to this research.	We have revised sections relating to PPI to improve coherence and readability (lines 216-230 and 334-393). As far as possible we have provided demographic information about public contributors but were not provided with exact ages or self-reported ethnicity.